# WILDFEEDBACK: ALIGNING LLMS WITH IN-SITU USER INTERACTIONS AND FEEDBACK

## ABSTRACT

As large language models (LLMs) continue to advance, aligning these models with human preferences has emerged as a critical challenge. Traditional alignment methods, relying on human or LLM annotated datasets, are limited by their resource-intensive nature, inherent subjectivity, misalignment with real-world user preferences, and the risk of feedback loops that amplify model biases. To overcome these limitations, we introduce WILDFEEDBACK, a novel framework that leverages in-situ user feedback during conversations with LLMs to create preference datasets automatically. Given a corpus of multi-turn user-LLM conversation, WILDFEEDBACK identifies and classifies user feedback to LLM responses between conversation turns. The user feedback is then used to create examples of preferred and dispreferred responses according to users' preference. Our experiments demonstrate that LLMs fine-tuned on WILDFEEDBACK dataset exhibit significantly improved alignment with user preferences, as evidenced by both traditional benchmarks and our proposed checklist-guided evaluation. By incorporating in-situ feedback from actual users, WILDFEEDBACK addresses the scalability, subjectivity, and bias challenges that plague existing approaches, marking a significant step toward developing LLMs that are more responsive to the diverse and evolving needs of their users.

## 1 INTRODUCTION

Large language models (LLMs) have become a cornerstone of modern natural language processing (NLP) applications, powering a wide range of tasks from conversational agents to content generation. Despite their strengths, aligning LLMs with human preferences remains a challenge (Bai et al., 2022a; Ouyang et al., 2022; OpenAI et al., 2024; Dubey et al., 2024). Traditional alignment methods involve instruction tuning and preference training on curated human or LLM-annotated datasets (Bai et al., 2022a; Ouyang et al., 2022; Cui et al., 2024). However, these approaches face critical limitations: human annotation is resource-intensive and often subjective, while LLM-generated synthetic data risks reinforcing biases instead of capturing diverse human preferences (Gautam & Srinath, 2024; Wyllie et al., 2024; Chen et al., 2024; Poddar et al., 2024).

In response, recent work explores in-situ user feedback (e.g., upvotes, downvotes, engagement) for LLM training Shi et al. (2022); Lin et al. (2024b); Don-Yehiya et al. (2024). This approach harnesses authentic user feedback during interactions with LLMs, offering a more dynamic and accurate reflection of user preferences. However, existing works are limited in scope. Shi et al. (2022) focus on explicit thumbs-up/thumbs-down style feedback. Lin et al. (2024b) and Don-Yehiya et al. (2024) move toward finer-grained utterance-level satisfaction estimation, but they treat each response in isolation and do not leverage the surrounding conversational context. As a result, these methods compress nuanced user reactions into narrowly scoped signals, missing the broader trajectory of user needs and expectations across turns. Moreover, prior approaches often fine-tune models directly on responses that trigger explicit feedback, without systematically capturing implicit feedback signals or the evolving dialogue state.

In this paper, we introduce WILDFEEDBACK, a novel framework designed to align LLMs with in-situ user interactions and feedback. WILDFEEDBACK addresses the limitations of existing approaches by constructing preference datasets from real user-LLM conversations, specifically focusing on user feedback that naturally occurs during these interactions. Unlike prior work, WILD-

054
055
056
057
058
059
060
061
062
063
064
065
066
067
068
069
070
071
072
073

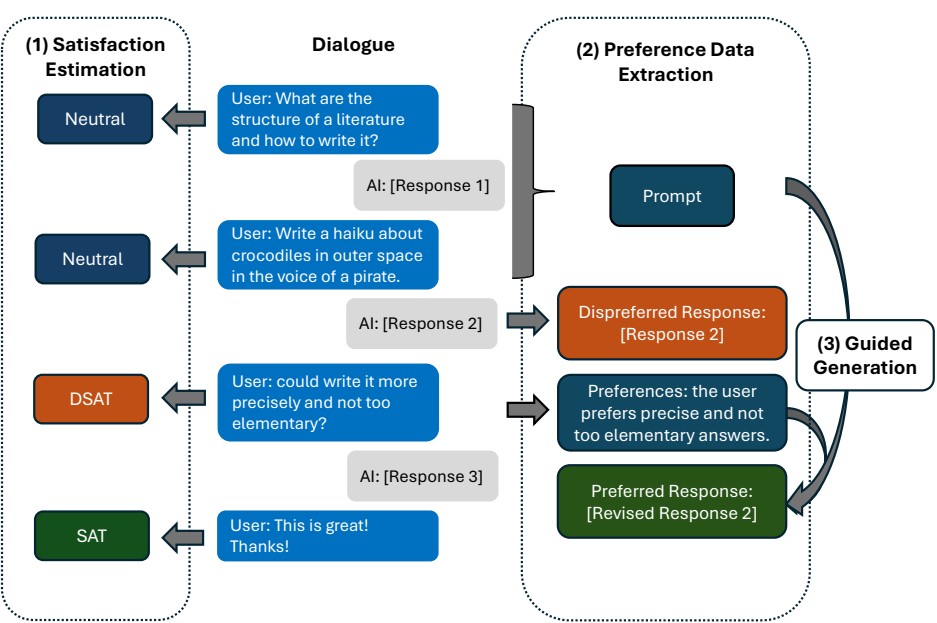

Figure 1: Overview of WILDFEEDBACK. (1) We begin by applying user satisfaction estimation to identify conversations and utterances that contain feedback signals. (2) We extract the entire conversation history leading up to a DSAT (dissatisfaction) signal as the prompt, and the response that triggers the DSAT as the dispreferred response. (3) Finally, we summarize the user's preferences based on the identified feedback signals and guide the generation of the preferred response

FEEDBACK explicitly leverages the full conversational history surrounding dissatisfaction signals, allowing us to infer preferences that are grounded in context rather than isolated utterances. The overview of the framework is shown in Figure 1. Our framework comprises three key components: (1) Feedback signal identification, which detects and classifies user feedback, distinguishing between positive and negative signals to infer user preferences; (2) Preference data construction, which transforms these signals into structured preference datasets; and (3) Checklist-guided evaluation, which systematically assesses model responses using an instance-level checklist derived from extracted user preferences as a rubric. This ensures that model improvements are grounded in real user expectations rather than predefined heuristics. To demonstrate the effectiveness of WILDFEED-BACK, we apply it to WildChat (Zhao et al., 2024), a dataset containing over 148,000 multi-turn conversations between users and ChatGPT (OpenAI et al., 2024) (see details of WildChat in Appendix E). This process results in a preference dataset of 20,281 samples[1], providing a rich resource for improving LLM alignment with real-world user preferences.

Through extensive experiments, we demonstrate that models fine-tuned on WILDFEEDBACK show significant improvements in aligning with user preferences, both in automated benchmarks and in our proposed checklist-guided evaluation framework. This work represents a step forward in creating more user-centric LLMs, with the potential to enhance user satisfaction across a wide range of applications. The contributions of this paper are threefold:

1. **In-situ User Preference Alignment**: we introduce WILDFEEDBACK, a novel framework that leverages naturally occurring user feedback in real conversations to ground LLM alignment in authentic, context-rich signals. By reflecting individual users' preferences, this approach mitigates the misalignment between external annotators and actual end-users.

2. **Scalable Preference Data Construction**: we adapt and extend user satisfaction estimation techniques to automatically identify both explicit and implicit feedback signals in multi-turn conversations. This process yields large, diverse, and fine-grained preference datasets across tasks, complementing the need for costly human annotation and making preference alignment both practical and scalable.

---

[1]The dataset will be released upon acceptance.

3. **Checklist-Guided Evaluation**: we propose a checklist-guided evaluation methodology that aligns the assessment of model performance with real user preferences, providing a more accurate benchmark for evaluating LLMs' alignment with human values.

## 2 RELATED WORK

**Feedback Learning for LLMs.** Incorporating human feedback has been shown to be an effective strategy to align LLMs with human preferences (Ouyang et al., 2022; Bai et al., 2022a; Dubey et al., 2024). However, relying human annotators to provide human feedback is inefficient and resource-intensive, which makes it hard to scale up. Additionally, human preferences are highly subjective. A small set of annotators may not represent broader preferences. Accordingly, some researchers aim to supervise AI models by model themselves (Bai et al., 2022b; Lee et al., 2023; Madaan et al., 2023; Burns et al., 2023; Li et al., 2023a). For instance, Bai et al. (2022b) introduced constitutional AI, in which they prompt LLMs to self-refine their own generations given a set of human-defined constitutions. However, relying on model's own feedback can create a feedback loop where the model's outputs increasingly reflect its own biases rather than diverse and authentic human perspectives. Recently, researchers have begun exploring the mining of user preferences from natural human-LLM interactions (Shi et al., 2022; Lin et al., 2024b; Don-Yehiya et al., 2024). These approaches capture real-time user feedback for more accurate preference alignment. Our work builds on this trend by leveraging in-situ user interactions to create preference datasets that better align with actual human values, addressing the limitations of both synthetic and human-annotated preference datasets.

**Data for LLM Alignment.** LLM alignment typically consists of two steps: instruction tuning and preference training. Instruction tuning, or supervised finetuning (SFT), aims to finetune models with a set of instruction-response pairs. Early works incorporated various NLP tasks for instruction tuning, demonstrating that LLMs could generalize well across different tasks (Wang et al., 2022; Chung et al., 2022; Ouyang et al., 2022). Subsequent research focused on constructing instruction data by directly distilling from capable LLMs (Wang et al., 2023; Xu et al., 2023). Researchers later recognized that preference training could further boost model performance across various tasks (Ouyang et al., 2022; Dubey et al., 2024). Preference training uses desired and undesired responses, either human-annotated (Bai et al., 2022a) or LLM-generated (Cui et al., 2024). Beyond general-purpose preference datasets, some datasets focus on specific tasks, such as summarization (Wu et al., 2021), model safety (Ji et al., 2023; Shi et al., 2024), and mathematics (Lightman et al., 2023). However, these approaches often rely on curated datasets that are either manually annotated by human experts or generated by models like GPT-4 (OpenAI et al., 2024). While these datasets provide a useful foundation, they may not fully capture the complexity and diversity of real-world user interactions. Our work addresses this gap by introducing a framework that leverages real-time feedback from actual users, allowing for more authentic and context-sensitive alignment of LLMs with true human preferences.

## 3 WILDFEEDBACK

Existing preference datasets often suffer from a mismatch between actual human preferences and those of the annotators (Chen et al., 2024; Poddar et al., 2024). Synthetic preference datasets, such as ULTRAFEEDBACK (Cui et al., 2024), rely solely on GPT-4 to generate rankings and determine which responses are preferred or dispreferred. However, this approach may not accurately capture real human values or nuanced preferences. Relying on synthetic data can create a feedback loop where the model's outputs increasingly reflect its own biases rather than diverse and authentic human perspectives. On the other hand, preference datasets annotated by human annotators are difficult to scale due to time and budget constraints (Bai et al., 2022a; Ouyang et al., 2022; Dubey et al., 2024). Moreover, human annotators' preferences can be highly subjective, often differing significantly from those of real users (Zhang et al., 2024; Fleisig et al., 2023).

To address these challenges, we introduce WILDFEEDBACK, a framework designed to align LLMs with in-situ user interactions and feedback. Unlike previous approaches that rely on synthetic responses, our framework directly learns preferences from real-world users, capturing both explicit and implicit feedback signals. The framework comprises three steps: (1) feedback signal identifica-

tion, (2) preference data construction, and (3) checklist-guided evaluation. The pipeline is illustrated in Figure 1. We apply this framework to WildChat (Zhao et al., 2024), a corpus of real user-ChatGPT conversations , and obtained the WILDFEEDBACK dataset, a preference dataset of 20,281 samples.

## 3.1 FEEDBACK SIGNALS IDENTIFICATION

To construct preference data from natural human-LLM interactions, we first identify conversations that contain feedback signals. This can be achieved through user satisfaction estimation. In multi-turn conversational sessions, a user may explicitly express their satisfaction (e.g., "thank you") or dissatisfaction (e.g., "revise it") in their utterances. Lin et al. (2024b) proposed a framework named SPUR that can automatically learn and identify SAT (satisfaction) and DSAT (dissatisfaction) patterns. SPUR generalizes SAT/DSAT rubrics from conversations with annotated thumb feedback by recursively prompting GPT-4. These rubrics can then be used to score a user's overall satisfaction or dissatisfaction, allowing us to identify utterances containing feedback signals.

WILDFEEDBACK adapts the SAT/DSAT rubrics from Lin et al. (2024b) with minor modifications. In total, we use 9 SAT and 9 DSAT rubrics. The SAT criteria include gratitude, learning, compliance, praise, personal details, humor, acknowledgment, positive closure, and getting there. The DSAT criteria consist of negative feedback, revision, factual error, unrealistic expectation, no engagement, ignored, lower quality, insufficient detail, and style. Detailed definitions of these rubrics can be found in Table 4 and 5. To streamline the process, we input these rubrics into GPT-4 [2] and prompt it to perform the classification at the utterance level. The complete prompt is available in the Appendix A.1. In total, there are 148,715 multi-turn conversations in the WildChat dataset, with approximately 12.8% of the multi-turn conversations containing feedback signals. Detailed statistics and analysis are presented in Table 1 and Section 5.2.

To ensure the reliability of GPT-4's classification of SAT/DSAT signals, we conducted a validation process using human expert annotators. Our findings indicate that GPT-4's ability to identify SAT/DSAT signals shows relatively high agreement with human annotations, achieving a Cohen's Kappa of $\kappa = 0.69$ for SAT and $\kappa = 0.50$ for DSAT, similar to the human performance. A detailed breakdown of GPT-4's performance and the human annotation process are provided in Appendix B.2.

Table 1: Statistics of SAT/DSAT in conversations. A conversation is labeled as SAT/DSAT if it contains at least one SAT/DSAT utterance.

| Category | SAT | DSAT | Total |
|---|---|---|---|
| # Conversations | 5,447 | 13,582 | 148,715 |
| # Utterances | 8,186 | 27,711 | 628,467 |

## 3.2 PREFERENCE PAIR GENERATION

After identifying conversations that contain feedback signals using the SAT/DSAT rubrics, we can construct semi-synthetic preference pairs. Each preference pair sample consists of four components: the prompt, user preferences, the preferred response, and the dispreferred response. For conversations with SAT/DSAT signals, we first analyze user responses marked by these signals and ask GPT-4 to summarize user preferences based on these feedback signals (e.g., the user prefers concise and direct answers). We then extract the conversation up to the model response that triggers the SAT/DSAT signals and use this as the prompt for our preference data.

For preferred and dispreferred response generation, we explore two different approaches: expert responses and on-policy responses. Specifically, we use GPT-4 for expert response generation, while Phi 3 (Abdin et al., 2024), Qwen 2 (Yang et al., 2024), and LLaMA 3 (Dubey et al., 2024) are employed for on-policy response generation. For expert responses, those that trigger DSAT signals in the original conversations are directly used as dispreferred responses (e.g., response 2 in Fig. 1). We then prompt GPT-4 to generate the preferred responses by using summarized user preferences as the system prompt. For on-policy responses, both preferred and dispreferred responses are generated by the policy model. The dispreferred responses are generated directly, whereas the preferred responses are produced using the summarized user preferences as the system prompt. Furthermore,

---

[2]Unless otherwise specified, in all of our experiments, we use GPT-4o with the `gpt-4o-0513` engine. For open-weight models, we use `Phi-3-mini-4k-instruct`, `Qwen2-7B-Instruct`, `Meta-Llama-3-8B-Instruct`.

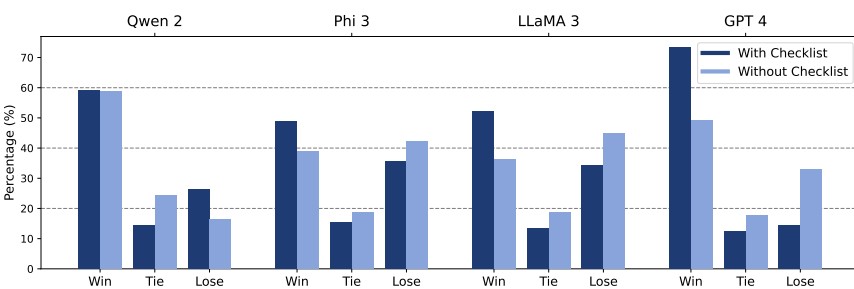

Figure 2: Comparison of in-situ user alignment across datasets generated by different models. "Win/Tie/Lose" represents the percentage of instances where the preferred responses win/tie/lose compared to the dispreferred responses in the WILDFEEDBACK dataset, prior to filtering. The comparison is made both with and without providing GPT-4 with summarized user preferences as checklists to guide its evaluation. With checklists, the preferred responses can be better distinguished.

recognizing that some user preferences may be harmful (e.g., preferences for explicit content), we take extra safety precautions. When prompting either the on-policy models or GPT-4 to generate preferred responses, we include an additional system instruction: "The response should be safe." Some conversations are also automatically filtered by the OpenAI moderation API. The prompt used for preference pair construction is provided in Appendix A.2.

### 3.3 CHECKLIST-GUIDED EVALUATION

Existing automated benchmarks, such as AlpacaEval (Dubois et al., 2024) and MT-Bench (Zheng et al., 2023b), heavily rely on using LLMs as judges. These benchmarks typically prompt models with a set of queries and then ask LLMs like GPT-4 or Claude (Anthropic, 2023) to provide a score or rank the responses of different models. This approach is problematic because it relies heavily on the internal knowledge of LLMs, which are known to be biased towards longer responses or responses generated by themselves (Liu et al., 2024b; Thakur et al., 2024). Additionally, there is a mismatch between the preferences of LLMs as judges and those of humans, leading to evaluations that do not accurately reflect user preferences. Furthermore, using human annotators to rank model responses based on their subjective experiences is also not ideal, as there can be a mismatch between annotators' preferences and actual user preferences.

In response, we propose checklist-guided evaluation, a general evaluation framework that more accurately reflects real user preferences. In our preference data construction module, we not only construct preference data from user-LLM interactions but also summarize user preferences expressed in natural language. These preferences, based on real users' textual feedback, can be used to align LLMs's evaluation more closely with real users' preferences. Instead of asking human annotators to directly rank model responses, we should ask them to rank those responses based on real users' preferences. When using LLMs as evaluators, we can provide an instance-level checklist to guide their assessments. Our evaluation framework is adapted from WILDBENCH (Lin et al., 2024a), which has been shown to correlate well with human judgement in ranking model performance as an automatic metric. We employ a pairwise evaluation strategy, where GPT-4 compares two different responses to determine which performs better on a given task, using an instance-level, preference-guided checklist to inform the comparison. This metric allows for straightforward comparisons among models, with easily interpretable win/lose rates as intermediate outcomes. The full prompt can be found in Appendix A.3.

Similar to feedback signal identification (§3.1), to ensure the reliability of GPT-4 on checklist-guided evaluation, we conducted a validation process using human expert annotators. We found GPT-4 achieves an human agreement of 57.14%, similar to the human-human agreement of 63.27%. A detailed breakdown of GPT-4's performance and the human annotation process are provided in Appendix C.

|  | # Conv. | Prompt Length | Response Length | Multi-Turn? | Feedback Type |
|---|---|---|---|---|---|
| WebGPT (Nakano et al., 2022) | 38,925 | 51 | 188 | ✗ | Human Annotators |
| Anthropic HH (Bai et al., 2022a) | 118,263 | 186 | 95 | ✗ | Human Annotators |
| OASST1 (Köpf et al., 2023) | 35,905 | 168 | 221 | ✓ | Human Annotators |
| HELPSTEER2 (Wang et al., 2024) | 20,324 | 713 | 1,492 | ✗ | Human Annotators |
| ULTRAFEEDBACK (Cui et al., 2024) | 61,135 | 159 | 256 | ✗ | GPT-4 |
| WILDFEEDBACK (ours) |  |  |  |  |  |
| ↪ GPT-4 | 20,281 | 929 | 440 |  |  |
| ↪ Qwen 2 | 11,509 | 1,057 | 541 | ✓ | In-situ Users |
| ↪ Phi 3 | 9,194 | 931 | 344 |  |  |
| ↪ LLaMA 3 | 10,659 | 982 | 376 |  |  |

Table 2: Statistics of existing preference datasets. Length refers to number of tokens. The responses of WILDFEEDBACK are either extracted from the original conversations or generated by GPT-4, Qwen 2, Phi 3, or LLaMA 3.

## 3.4 WILDFEEDBACK DATA CONSTRUCTION

The preference pair construction approach described in Section 3.2 allows us to build a robust dataset for training models to better align responses with user preferences.

To evaluate whether our generated preferred responses align with actual user preferences, we randomly selected 500 samples from the WILDFEEDBACK datasets and performed checklist-guided evaluation (§3.3), comparing the preferred and dispreferred responses. As explained in Section 3.2, there are two versions of WILDFEEDBACK preference pairs: the GPT-4 version and the on-policy version, which differ in whether the responses are generated by GPT-4 or the policy model. As shown in Figure 2, we found that without checklist-guided evaluation, GPT-4 does not necessarily favor responses aligned with summarized user preferences, often defaulting to models' zero-shot generations instead. However, after providing the preferences as checklists to guide the evaluation, GPT-4's selections more closely align with real users' preferences. Additionally, we observed that GPT-4 is significantly more steerable than smaller models: over 70% of its preferred responses align with in-situ user preferences, compared to only about 50% for smaller models.

Since policy models are less steerable than GPT-4 and may not always align with provided user preferences, we apply an additional filtering process, discarding any on-policy pairs that do not align with user preferences based on checklist-guided evaluation. In contrast, we retain all GPT-4-generated preference pairs, as they consistently demonstrate higher alignment.

Table 2 reports statistics on WILDFEEDBACK constructed datasets compared with open-source datasets[3]. To the best of our knowledge, WILDFEEDBACK is the first multi-turn pairwise preference dataset derived from real human-LLM interactions. Unlike datasets annotated by human labelers or LLMs, which often fail to fully capture real user preferences, WILDFEEDBACK is built from in-situ user feedback. Although OpenAssistant Conversations (OASST1) (Köpf et al., 2023) also includes multi-turn conversations, its prompts and responses are fully composed by human annotators, making it less reflective of genuine human-LLM interactions. In the next section, we demonstrate that WILDFEEDBACK more accurately represents authentic human-LLM interactions, making it a more reliable resource for developing and evaluating preference-based models.

## 4 EXPERIMENT

To validate the effectiveness of WILDFEEDBACK, we finetune models from different families on it and compare their performances with the vanilla models and the models finetuned on ULTRA-FEEDBACK data. We evaluate models' performance on general benchmarks and a held-out test set of WILDFEEDBACK using checklist-guided evaluation.

---

[3]For ULTRAFEEDBACK, we refer to the pre-processed, binarized version used to train Zephyr (Tunstall et al., 2023).

**Models and training settings.** We use off-the-shelf instruction-tuned Qwen 2, Phi 3, and LLaMA 3 models. As described in Section 3.2, each model is fine-tuned on two versions of both WILD-FEEDBACK (WF) and ULTRAFEEDBACK (UF): a GPT-4 version and an on-policy version.

For WILDFEEDBACK, the WF GPT-4 setup utilizes GPT-4 to generate preferred responses based on summarized user preferences. Dispreferred responses are extracted from conversations that contain DSAT signals. In the WF On-policy setup, each policy model (Qwen 2, Phi 3, or LLaMA 3) generates both preferred and dispreferred responses, again making use of summarized user preferences to produce the preferred ones. We train each model for one epoch of supervised fine-tuning (SFT) on the preferred responses, followed by one epoch of direct preference optimization (DPO) (Rafailov et al., 2023) on the entire dataset. We find that hyperparameter tuning is essential for optimal results (see Appendix D).

We also fine-tune models using ULTRAFEEDBACK, one of the most widely used preference datasets due to its superior performance compared to others. Models such as the Tulu 3 series Lambert et al. (2025) and Zephyr Tunstall et al. (2023) have been fine-tuned on this dataset. The prompts in UL-TRAFEEDBACK are sourced from various instruction datasets. Each prompt has four responses from different LLMs, numerically rated by GPT-4. However, due to the off-policy nature of ULTRA-FEEDBACK and the outdated models used to generate its responses, it has become common practice to regenerate responses using only the original prompts when training new models on this dataset (Meng et al., 2024; Dong et al., 2024; Xiong et al., 2024). Following this approach, we create two versions of the dataset: UF GPT-4 and UF On-policy. In UF GPT-4, we randomly select 20,000 prompts from ULTRAFEEDBACK, and GPT-4 generates two responses for each prompt. GPT-4 then acts as a judge, selecting the better response as the preferred one while marking the other as dispreferred. In UF On-policy, each policy model generates five responses per prompt, after which a GPT-4 judge selects the best response as preferred, while one of the remaining four is randomly designated as dispreferred. The specific prompt used to guide GPT-4 in selecting the preferred response is provided in Appendix A.4. By regenerating the responses for ULTRAFEEDBACK, we also ensure a fair comparison to our WILDFEEDBACK setup.

In summary, for all three policy models, we compare five configurations: (1) the off-the-shelf instruction-tuned model, (2) WF GPT-4, (3) WF On-policy, (4) UF GPT-4, and (5) UF On-policy.

**Benchmarks Evaluation.** We evaluate our models using three of the most popular open-ended instruction-following benchmarks: AlpacaEval 2 (Li et al., 2023b), MT-Bench (Zheng et al., 2023a), and Arena-Hard (Li et al., 2024). AlpacaEval 2 consists of 805 questions from 5 datasets, and MT-Bench covers 8 categories with 80 questions. Arena-Hard is an enhanced version of MT-Bench, incorporating 500 well-defined technical problem-solving queries. We report scores following each benchmark's evaluation protocol: For AlpacaEval 2, we report both the raw win rate (WR) and the length-controlled win rate (LC) (Dubois et al., 2024). The LC metric is specifically designed to be robust against model verbosity. For MT-Bench, we report the average MT-Bench score with GPT-4o (`gpt-4o-0513`) as the judge. For Arena-Hard, we report the win rate (WR) against the baseline model. As specified by the benchmarks, we use GPT-4-Turbo (`gpt-4-0125`) as the judge for both AlpacaEval 2 and Arena-Hard. We use the same, default decoding strategy specified by each evaluation benchmark respectively.

**WILDFEEDBACK Evaluation.** In addition to publicly available benchmarks, we constructed our own evaluation benchmark from the held-out test set in WILDFEEDBACK and evaluated models using checklist-guided evaluation (§3.3). We ensured that all test samples came from conversations and users that were never included in the training set. Constructing an evaluation dataset for checklist-guided evaluation is non-trivial, as we can no longer randomly or stratifiedly select test samples from different domains. In checklist-guided evaluation, we always provide a user-inspired checklist for GPT-4 to guide its evaluation, making it more aligned with real users' preferences. However, individual user preferences can be highly subjective and specific. The goal of WILDFEEDBACK is not to align language models with the preferences of a specific individual but to learn the broader mode of all individuals' preferences. Therefore, we must ensure that the preferences reflected in the test samples represent the majority view. Additionally, since the user preferences we extracted are often particular to specific tasks, we also need to ensure that the tasks in the test set are at least somewhat similar to those in the training set.

To achieve this, we utilized FAISS (Douze et al., 2024) to cluster user prompts and their summarized preferences. We grouped all user prompts into 70 clusters. Within each cluster, we selected 10 samples where the preferences were most similar to the other preferences in the same group. We then applied similar data curation techniques as described in WILDBENCH (Lin et al., 2024a) to perform deduplication and remove nonsensical tasks, resulting in a final test set of 540 samples. By doing so, we aim to provide a more reliable and comprehensive evaluation that reflects the majority's preferences without overfitting to specific, idiosyncratic cases.

For WILDFEEDBACK evaluation, we report the win, tie, lose percentage against the instruct models and the models trained on ULTRAFEEDBACK with GPT-4 as the judge. We employ the WILD-BENCH prompt (Lin et al., 2024a) to perform the evaluation, which has been shown to correlate well with human judgement in ranking model performance. We report the results evaluated with or without the user preferences provided as a checklist.

## 5 RESULTS AND DISCUSSIONS

### 5.1 MODEL PERFORMANCE

**Training models on WILDFEEDBACK significantly and consistently enhances performance across all benchmarks.** As shown in Table 3, models trained on either version of WILDFEED-BACK achieve higher performance across AlpacaEval 2, Arena-Hard, and MT-Bench. For example, after training on the GPT-4 version of WILDFEEDBACK (WF GPT-4), Phi 3's length-controlled win rate on AlpacaEval 2 increases from 24.3% to 34.9%, while its win rate on Arena-Hard improves from 15.4% to 32.4%. Similarly, its performance on MT-Bench rises from a score of 7.32 to 7.75. Models trained on WILDFEEDBACK also consistently outperform those on ULTRAFEEDBACK.

**WILDFEEDBACK significantly enhances model alignment with in-situ user feedback.** As detailed in Section §4, the WILDFEEDBACK test set is sourced from real human-ChatGPT conversations where users explicitly express dissatisfaction, implicitly suggesting that the models are poorly aligned with real user preferences on these tasks. As shown in Figure 3, models trained on either version of WILDFEEDBACK exhibit stronger alignment with real user preferences. For instance, LLaMA 3 trained on WF GPT-4 outperforms the LLaMA 3 model trained on ULTRAFEEDBACK 45.5% of the time, while losing only 38.8% of the time when evaluated without a checklist. When real user preferences are provided as checklists to guide GPT-4's evaluation, the win rate further increases to 50.8%, highlighting that models trained on WILDFEEDBACK better align with actual user preferences compared to the off-the-shelf models and those trained on ULTRAFEEDBACK.

Table 3: AlpacaEval 2, Arena-Hard, and MT-Bench results under the four settings. LC and WR denote length-controlled and raw win rate. WF/UF On-policy/GPT-4 refers to the model trained on the on-policy/GPT-4 version of WILDFEED-BACK/ULTRAFEEDBACK.

| Models | AlpacaEval 2 | | Arena-Hard | MT-Bench |
|---|---|---|---|---|
| | LC (%) | WR (%) | WR (%) | Score |
| Phi 3 | 24.3 | 17.4 | 15.4 | 7.32 |
| ↪ WF On-Policy | 29.0 | 27.1 | 30.1 | 7.42 |
| ↪ UF On-Policy | 27.2 | 25.9 | 28.7 | 7.40 |
| ↪ WF GPT-4 | 34.9 | 36.6 | 32.4 | 7.75 |
| ↪ UF GPT-4 | 32.5 | 38.4 | 30.5 | 7.68 |
| LLaMA 3 | 22.9 | 22.6 | 20.6 | 7.10 |
| ↪ WF On-Policy | 30.1 | 29.6 | 22.1 | 7.15 |
| ↪ UF On-Policy | 28.8 | 34.1 | 20.2 | 7.04 |
| ↪ WF GPT-4 | 34.2 | 42.8 | 32.9 | 7.57 |
| ↪ UF GPT-4 | 32.2 | 43.2 | 32.6 | 7.49 |
| Qwen 2 | 28.7 | 26.0 | 24.9 | 7.55 |
| ↪ WF On-Policy | 42.6 | 34.4 | 36.1 | 8.02 |
| ↪ UF On-Policy | 38.3 | 34.2 | 29.2 | 7.72 |
| ↪ WF GPT-4 | 39.4 | 33.5 | 27.9 | 7.60 |
| ↪ UF GPT-4 | 40.6 | 32.5 | 27.6 | 7.66 |

### 5.2 A DEEPER DIVE INTO USER'S FEEDBACK TYPES

In addition to improving model performance, WILDFEEDBACK also provides a lens to diagnose and interpret user feedback, unlike previous benchmarks that only offer a scalar score. To better understand how different types of user feedback surface in practice, we also instruct expert annotators to provide justification to binary SAT/DSAT annotation based on our rubrics (see Table 4 and Table 5). The resulting distributions are summarized in Figure 4. Dissatisfaction was most often linked

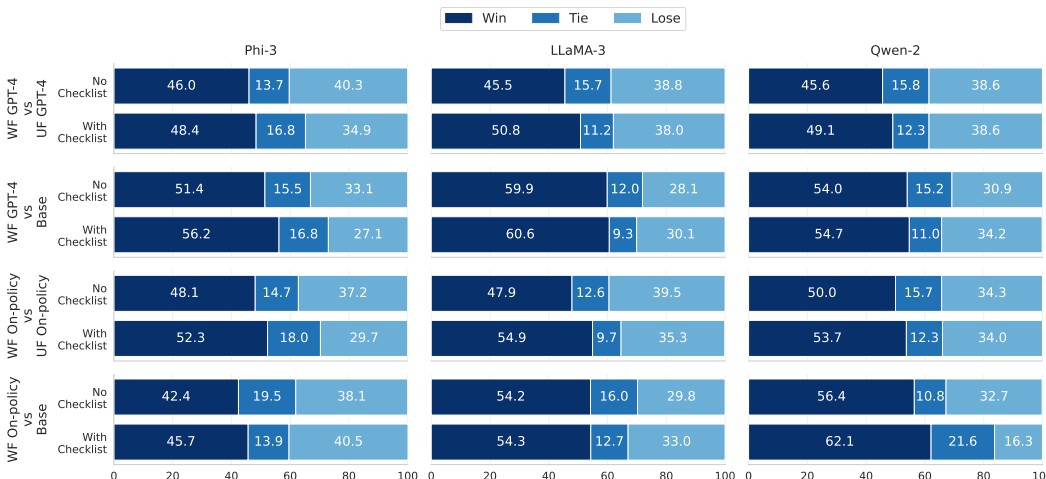

Figure 3: Preference evaluation on the WILDFEEDBACK test set, with or without the checklist. All numbers are the percentages of win/tie/lose. WF/UF On-policy/GPT-4 refers to the model trained on the on-policy/GPT-4 version of WILDFEEDBACK/ULTRAFEEDBACK. Base models here refers to the off-the-shelf instruct models. Models trained on WILDFEEDBACK consistently outperformed all the baselines.

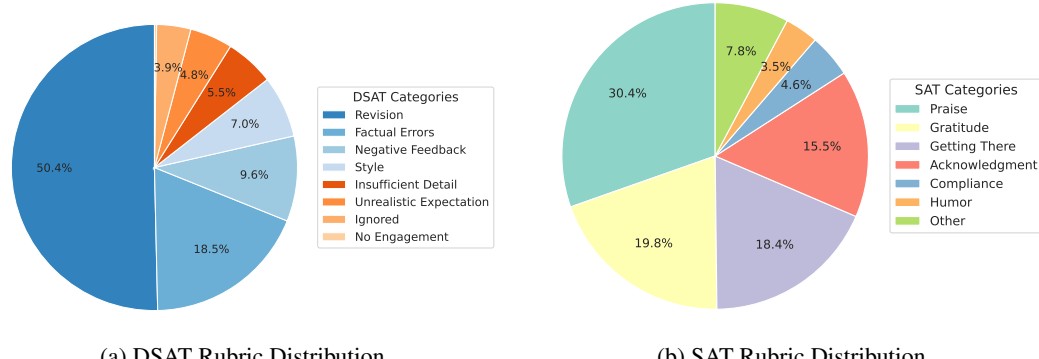

(a) DSAT Rubric Distribution.  (b) SAT Rubric Distribution.

Figure 4: Comparison of rubric distributions for DSAT and SAT categories.

to revision needs or factual inaccuracies, while more subtle signals such as style appeared less frequently. By contrast, satisfaction was expressed across a more diverse set of categories, including praise, gratitude, and acknowledgment of progress. Overall, these findings suggest that dissatisfaction is dominated by concrete issues of factuality and revision, whereas satisfaction arises from a broader set of positive responses such as praise, gratitude, and recognition of progress. A more detailed breakdown of annotation procedures and additional analysis of category-level differences are provided in Appendix B.2.

## 6 CONCLUSION

In this work, we propose a framework for constructing preference data and evaluating conversational AI models based on natural human-LLM interactions. By using SAT/DSAT rubrics to identify user satisfaction and dissatisfaction in conversations, we create a preference dataset that includes user prompts, preferences, and both preferred and dispreferred responses. This enables models to better align with user expectations. Additionally, we introduce a checklist-guided evaluation framework that addresses biases in existing benchmarks by using real user feedback to guide LLM evaluations, ensuring a more accurate reflection of user preferences. Our method aligns LLMs with diverse human values, enhancing user satisfaction.

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

# A PROMPTS

## A.1 PROMPT FOR FEEDBACK SIGNALS IDENTIFICATION

The following is the full prompt we used for dialogue state tracking and SAT/DSAT classification. In addition, we also prompt GPT-4 to do domain and intent classification. The prompt is adapted from Das et al. (2023) and Lin et al. (2024b).

```
## LABEL DEFINITION ##
{
"valid_preceding_topical_relation_labels": [
{
"label": "YES",
"definition":                  "The current turn has **some or any**
topical/subtopical relation to the preceding conversation
context."
},
{
"label": "NO",
"definition":                  "The current turn has **absolutely no**
topical/subtopical relation to the preceding conversation context
OR is the first turn in the conversation, marking the beginning of
a new dialogue segment."
}
],
"valid_domain_labels": [
"AI MACHINE LEARNING AND DATA SCIENCE",
"ASTROLOGY",
"BIOLOGY AND LIFE SCIENCE",
"BUSINESS AND MARKETING",
"CAREER AND JOB APPLICATION",
"CLOTHING AND FASHION",
"COOKING FOOD AND DRINKS",
"CRAFTS",
"CULTURE AND HISTORY",
"CYBERSECURITY",
"DATING FRIENDSHIPS AND RELATIONSHIPS",
"DESIGN",
"EDUCATION",
"ENTERTAINMENT",
"ENVIRONMENT AGRICULTURE AND ENERGY",
"FAMILY PARENTING AND WEDDINGS",
"FINANCE AND ECONOMICS",
"GAMES",
"GEOGRAPHY AND GEOLOGY",
"HEALTH AND MEDICINE",
"HOUSING AND HOMES",
"HUMOR AND SARCASM",
"LANGUAGE",
"LAW AND POLITICS",
"LITERATURE AND POETRY",
"MANUFACTURING AND MATERIALS",
"MATH LOGIC AND STATISTICS",
"MUSIC AND AUDIO",
"NEWS",
"PETS AND ANIMALS",
"PHILOSOPHY",
"PHYSICS CHEMISTRY AND ASTRONOMY",
"PRODUCTIVITY",
```

```
972    "PSYCHOLOGY AND EMOTIONS",
973    "RELIGION AND MYTHOLOGY",
974    "SHIPPING AND DELIVERY",
975    "SHOPPING AND GIFTS",
976    "SMALL TALK",
977    "SOCIAL MEDIA",
978    "SOFTWARE AND WEB DEVELOPMENT",
979    "SPORTS AND FITNESS",
980    "TAXATION",
981    "TECHNOLOGY",
982    "TIME AND DATES",
983    "TRANSPORTATION AUTOMOTIVE AND AEROSPACE",
984    "TRAVEL",
985    "VISUAL ARTS AND PHOTOGRAPHY",
986    "WEATHER",
987    "WRITING JOURNALISM AND PUBLISHING",
988    "OTHER"
989    ],
990    "valid_intent_labels":[
991    {
992    "label": "INTENT:1-INFORMATION_SEEKING",
993    "definition":       "The user wants to find factual information or
994    answers to specific questions."
995    },
996    {
997    "label": "INTENT:2-ANALYSIS",
998    "definition":    "The user asks analytical or conceptual questions
999    about a complex topic or problem.  The user's questions require
1000   some degree of reasoning, interpretation, argumentation,
1001   comparison, and/or data processing."
1002   },
1003   {
1004   "label": "INTENT:3-CREATION",
1005   "definition":  "The user asks the agent to either generate original
1006   content or translate existing content into new content based on
1007   specified criteria or constraints."
1008   },
1009   {
1010   "label": "INTENT:4-OPEN-ENDED_DISCOVERY",
1011   "definition":     "The user wants to casually chat or play with the
1012   agent out of curiosity, boredom, or humor, OR the user's intent
1013   is so unclear/underspecified that it's impossible to categorize
1014   in any of the other intent classes.  The user mainly treats the
1015   agent as a conversation or chitchat partner, and none of the other
1016   intent categories can be assigned."
1017   }
1018   ],
1019   "valid_satisfaction_labels":[
1020   {
1021   "label": "Gratitude",
1022   "definition":  "The user thanks or compliments the AI agent for its
1023   responses"
1024   },
1025   {
       "label": "Learning",
       "definition":               "The user learns something new or useful by
       indicating curiosity and satisfaction with the information
       provided"
       },
```

```
1026   {
1027   "label": "Compliance",
1028   "definition":        "The user follows the AI agent's suggestions or
1029   instructions when applicable"
1030   },
1031   {
1032   "label": "Praise",
1033   "definition":         "The user uses positive feedback words (e.g.,
1034   excellent, amazing) or emojis, indicating enthusiasm and enjoyment
1035   of the conversation"
1036   },
1037   {
1038   "label": "Personal_Details",
1039   "definition":    "The user shares more personal details or opinions
1040   with the AI agent when satisfied with its responses"
1041   },
1042   {
1043   "label": "Humor",
1044   "definition": "The user jokes with or challenges the AI agent in a
1045   friendly manner when suitable"
1046   },
1047   {
1048   "label": "Acknowledgment",
1049   "definition":          "The user acknowledges or confirms that they
1050   understood or agreed with the AI agent's explanations when
1051   relevant"
1052   },
1053   {
1054   "label": "Positive_Closure",
1055   "definition":    "The user ends the conversation on a positive note
1056   without asking for more information or assistance"
1057   },
1058   {
1059   "label": "Getting_There",
1060   "definition":        "The user acknowledges that the model's response
1061   is getting better or has merit but is not fully satisfied.
1062   Appropriate dissatisfaction criteria may need to be checked as
1063   well when Getting_There presents"
1064   },
1065   {
1066   "label": "N/A",
1067   "definition":     "The user utterance of the turn does NOT match the
1068   definition of any other valid satisfaction labels"
1069   }
1070   ],
1071   "valid_dissatisfaction_labels":[
1072   {
1073   "label": "Negative_Feedback",
1074   "definition":          "The user explicitly expresses dissatisfaction,
1075   frustration, annoyance, or anger with the AI agent's response or
1076   behavior"
1077   },
1078   {
1079   "label": "Revision",
       "definition": "The user explicitly asks the AI agent to revise its
       previous response or repeatedly asks similar questions"
       },
       {
       "label": "Factual_Error",
```

```
1080   "definition": "The user points out the AI agent's factual mistakes,
1081   inaccuracies, or self-contradiction in its information or output"
1082   },
1083   {
1084   "label": "Unrealistic_Expectation",
1085   "definition": "The user has unrealistic expectations of what the AI
1086   agent can do and does not accept its limitations or alternatives"
1087   },
1088   {
1089   "label": "No_Engagement",
1090   "definition":          "The user does not respond to the AI agent's
1091   questions, suggestions, feedback requests, etc."
1092   },
1093   {
1094   "label": "Ignored",
1095   "definition":          "The user implies that their query was ignored
1096   completely or that the response did not address their intent/goal
1097   at all"
1098   },
1099   {
1100   "label": "Lower_Quality",
1101   "definition":   "The user perceives a decline in quality of service
1102   compared to previous experience with other agents/tools, etc."
1103   },
1104   {
1105   "label": "Insufficient_Detail",
1106   "definition": "The user wants more specific/useful information than
1107   what is provided by the AI agent"
1108   },
1109   {
1110   "label": "Style",
1111   "definition":      "The user feels that there is a mismatch between
1112   their preferred style (e.g.  bullet point vs paragraph, formal
1113   vs casual, short vs long, etc.)  and what is provided by the AI
1114   agent"
1115   },
1116   {
1117   "label": "N/A",
1118   "definition":   "The user utterance of the turn does NOT match the
1119   definition of any other valid dissatisfaction labels"
1120   }
1121   ],
1122   "valid_state_labels": [
1123   {
1124   "label": "FEEDBACK",
1125   "definition": "The user utterance of the turn contains a comment or
1126   evaluation or judgement of the previous turn's agent response"
1127   },
1128   {
1129   "label": "REFINEMENT",
1130   "definition":     "The user utterance of the turn is a repetition or
1131   refinement of unclear/underspecified instruction given in the
1132   previous turn's user utterance"
1133   },
1134   {
1135   "label": "NEWTOPIC",
1136   "definition":    "The user utterance of the turn is either the first
1137   turn of the conversation or is not related in terms of topic or
1138   task to its previous turn, introducing a new topic or task"
```

```
},
{
"label": "CONTINUATION",
"definition":       "The user utterance of the turn is a topical or
logical continuation of the previous turn"
}
]
}
```

## TASK ##
You are given a dialogue between a user and an agent comprised of turns starting with T. For each turn, solely based on the turn's User utterance, you must carefully analyze the conversation and answer the following questions by replacing $instruction$ with correct answers in JSON format. - Summarize the user utterance in $\leq 3$ sentences
- Analyze the user utterance's relation with the previous turn and output an appropriate label from the "valid_preceding_topical_relation_labels" list.
- Analyze the user utterance's domain and output an appropriate label from the "valid_domain_labels" list.  If preceding_topical_relation is YES, the domain label must be consistent with the preceding turn's domain label.
- Analyze the user utterance's intent and output an appropriate label from the "valid_intent_labels" list.
- Analyze the user utterance's satisfaction with respect to the previous turn's AI response and output all applicable labels from the "valid_satisfaction_labels" list.
- Analyze the user utterance's dissatisfaction with respect to the previous turn's AI response and output all applicable labels from the "valid_dissatisfaction_labels" list.
- Analyze the user utterance's state and output an appropriate label from the "valid_state_labels" list.

## OUTPUT FORMAT ##
The length and turn order of the output list must match the length and turn order of the input list. The sample output format is given as follow: [ {
```
"T-$turn number$": {
"summary": "$turn summary in ≤ 3 sentence$",
"preceding_topical_relation":       "$an appropriate valid preceding
topical relation label$",
"domain": "$an appropriate valid domain label$",
"intent": "INTENT:$an appropriate valid intent label$",
"satisfaction": [$a comma separated string list of applicable valid
satisfaction label(s)$],
"dissatisfaction":   [$a comma separated string list of applicable
valid dissatisfaction label(s)$],
"state": "$an appropriate valid state label$"
}
}]
```

## INPUT ##
#D1#

## OUTPUT ##

A.2   PROMPT FOR PREFERENCE PAIR CONSTRUCTION

The following is the prompt for constructing preference data.

# Conversation between User and AI
<|begin_of_history|>
history
<|end_of_history|>

# Instruction
What are the user's query and preferences? The query should be the user's first attempt before providing any feedbacks to the model. Only output the turn id. The preference should always be based on user's feedbacks and in complete sentences. Generate your answer in json format like

```
[{
"query":  turn id,
"preferences":  [preference 1, preference 2, ...]
}]
```

## A.3 PROMPT FOR CHECKLIST-GUIDED EVALUATION

The following is the prompt for checklist-guided evaluation. We borrow the WB-Reward prompt from WILDBENCH (Lin et al., 2024a).

# Instruction
You are an expert evaluator. Your task is to evaluate the quality of the responses generated by two AI models. We will provide you with the user query and a pair of AI-generated responses (Response A and B). You should first read the user query and the conversation history carefully for analyzing the task, and then evaluate the quality of the responses based on and rules provided below.
# Conversation between User and AI
## History
<|begin_of_history|>
{history}
<|end_of_history|>
## Current User Query
<|begin_of_query|>
{query}
<|end_of_query|>
## Response A
<|begin_of_response_A|>
{response_a}
<|end_of_response_A|>
## Response B
<|begin_of_response_B|>
{response_b}
<|end_of_response_B|>
# Evaluation
## Checklist
<|begin_of_checklist|>
{checklist}
<|end_of_checklist|>
Please use this checklist to guide your evaluation, but do not limit your assessment to the checklist.
## Rules
You should compare the above two responses based on your analysis of the user queries and the conversation history. You should first write down your analysis and the checklist that you used for the evaluation, and then provide your assessment according to the checklist. There are five choices to give your final assessment: ["A++", "A+", "A=B", "B+", "B++"], which correspond to the following meanings:
- 'A++': Response A is much better than Response B.
- 'A+': Response A is only slightly better than Response B.
- 'A=B': Response A and B are of the same quality. Please use this choice sparingly.
- 'B+': Response B is only slightly better than Response A.
- 'B++': Response B is much better than Response A.
## Output Format
First, please output your analysis for each model response, and then summarize your assessment to three aspects: "reason A=B", "reason A > B", and "reason B > A", and finally make your choice for the final assessment. Please provide your evaluation results in the following json format by filling in the placeholders in []:

```
{
"analysis of A":  "[analysis of Response A]",
"analysis of B":  "[analysis of Response B]",
"reason of A=B":  "[where Response A and B perform equally well]",
"reason of A>B":  "[where Response A is better than Response B]",
"reason of B>A":  "[where Response B is better than Response A]",
"choice":  "[A++ or A+ or A=B or B+ or B++]"
}
```

## A.4 PROMPT FOR DATASET EVALUATION

The following is the prompt for constructing the on-policy version of the ULTRAFEEDBACK dataset. The prompt is adapted from the WB-Reward prompt (Lin et al., 2024a).

# Instruction
You are an expert evaluator. Your task is to evaluate the quality of the responses generated by two AI models. We will provide you with the user query and a set of AI-generated responses (Response A, Response B, Response C, Response D, Response E). You should first read the user query and the conversation history carefully for analyzing the task, and then evaluate the quality of the responses based on the rules provided below.
# Conversation between User and AI
## History
<|begin_of_history|>
{history}
<|end_of_history|>
## Current User Query
<|begin_of_query|>
{query}
<|end_of_query|>
## Response A
<|begin_of_response_A|>
{response_a}
<|end_of_response_A|>
## Response B
<|begin_of_response_B|>
{response_b}
<|end_of_response_B|>
## Response C
<|begin_of_response_C|>
{response_c}
<|end_of_response_C|>
## Response D
<|begin_of_response_D|>
{response_d}
<|end_of_response_D|>
## Response E
<|begin_of_response_E|>
{response_e}
<|end_of_response_E|>
# Evaluation
## Checklist
<|begin_of_checklist|>
{checklist}
<|end_of_checklist|>
Please use this checklist to guide your evaluation, but do not limit your assessment to the checklist.
## Rules
You should compare the above five responses based on your analysis of the user queries and the conversation history. You should first write down your analysis and the checklist that you used for the evaluation, and then provide your assessment according to the checklist.

There are six choices to give your final assessment: ["A", "B", "C", "D", "E", "A=B=C=D=E"], which correspond to the following meanings:
- 'A': Response A is much better than the other responses.
- 'B': Response B is much better than the other responses.
- 'C': Response C is much better than the other responses.
- 'D': Response D is much better than the other responses.
- 'E': Response E is much better than the other responses.
- 'A=B=C=D=E': Response A, B, C, D, E are of the same quality. No response particularly stood out. Please use this choice sparingly.
## Output Format
First, please output your analysis for each model response, and then summarize your assessment to "comparison of A, B, C, D, E", and finally make your choice for the final assessment. Please provide your evaluation results in the following json format by filling in the placeholders in []:

```
{
"analysis of A":  "[analysis of Response A]",
"analysis of B":  "[analysis of Response B]",
"analysis of C":  "[analysis of Response C]",
"analysis of D":  "[analysis of Response D]",
"analysis of E":  "[analysis of Response E]",
"comparison of A, B, C, D, E":  "[where Response A, B, C, D, E
perform equally well]",
"choice":  "[A or B or C or D or E or A=B=C=D=E]"
}
```

# B    SAT AND DSAT

## B.1    DETAILED SAT AND DSAT CRITERIA

The detailed definitions of SAT and DSAT can be found in Table 4 and Table 5.

| Keyword | Definition |
|---|---|
| Gratitude | The user thanks or compliments the AI agent for its responses. |
| Learning | The user learns something new or useful by indicating curiosity and satisfaction with the information provided. |
| Compliance | The user follows the AI agent's suggestions or instructions when applicable. |
| Praise | The user uses positive feedback words (e.g., excellent, amazing) or emojis, indicating enthusiasm and enjoyment of the conversation. |
| Personal Details | The user shares more personal details or opinions with the AI agent when satisfied with its responses. |
| Humor | The user jokes with or challenges the AI agent in a friendly manner when suitable. |
| Acknowledgment | The user acknowledges or confirms that they understood or agreed with the AI agent's explanations when relevant. |
| Positive Closure | The user ends the conversation on a positive note without asking for more information or assistance. |
| Getting There | The user acknowledges that the model's response is getting better or has merit but is not fully satisfied. |

Table 4: Detailed definitions of the SAT Rubrics.

| Keyword | Definition |
|---------|------------|
| Negative Feedback | The user explicitly expresses dissatisfaction, frustration, annoyance, or anger with the AI agent's response or behavior. |
| Revision | The user explicitly asks the AI agent to revise its previous response or repeatedly asks similar questions. |
| Factual Error | The user points out the AI agent's factual mistakes, inaccuracies, or self-contradiction in its information or output. |
| Unrealistic Expectation | The user has unrealistic expectations of what the AI agent can do and does not accept its limitations or alternatives. |
| No Engagement | The user does not respond to the AI agent's questions, suggestions, feedback requests, etc. |
| Ignored | The user implies that their query was ignored completely or that the response did not address their intent/goal at all. |
| Lower Quality | The user perceives a decline in quality of service compared to previous experience with other agents/tools, etc. |
| Insufficient Detail | The user wants more specific/useful information than what is provided by the AI agent. |
| Style | The user feels that there is a mismatch between their preferred style and what is provided by the AI agent. |

Table 5: Detailed definitions of the DSAT Rubrics.

## B.2 SAT AND DSAT ANNOTATION

**Human-ChatGPT Agreements.** We randomly sampled 50 multi-turn conversations, totaling over 500 utterances, and assigned 4 expert annotators to perform the same classification task. Each conversation was annotated by at least 2 annotators, resulting in a final Cohen's Kappa agreement of $\kappa = 0.70$ for SAT and $\kappa = 0.54$ for DSAT. For human annotation, we utilized a web-based annotation tool named Potato (Pei et al., 2022). The interface is shown in Figure 5. After completing the annotations, the annotators reviewed and discussed any disagreements, resolving conflicts to establish a ground truth test set of 50 conversations. GPT-4's performances on SAT and DSAT classification can be found in table 8. GPT-4 demonstrates strong performance in classifying SAT (satisfaction) signals, with high accuracy at 91.7% and balanced precision and recall, both around 73%. The Cohen's Kappa of 68.5% reflects substantial agreement with human annotators. For DSAT (dissatisfaction) signals, GPT-4 achieves a precision of 83.3%, with a recall of 48.4%, leading to an F1 score of 61.2% and a Cohen's Kappa of 50.4%. These metrics indicate that GPT-4 is effective at recognizing both SAT and DSAT signals.

**SAT/DSAT Distributions.** As depicted in Figure 5, in addition to binary SAT/DSAT classification, annotators were instructed to provide justifications based on rubric definitions, which are outlined in Table 5 and Table 4. The DSAT distribution in Table 6 shows that the most common category was Revision (50.36%), followed by Factual Errors (18.55%), Negative Feedback (9.64%), and Style (6.99%). Smaller shares were attributed to Insufficient Detail (5.54%), Unrealistic Expectation (4.82%), Ignored (3.86%), and No Engagement (0.24%). This indicates that dissatisfaction is dominated by revision needs and factual inaccuracies, while issues such as unmet expectations or lack of engagement appear less frequently. The SAT distribution in Table 7 is more evenly spread across categories, with Praise (30.39%), Gratitude (19.79%), Getting There (18.37%), and Acknowledgment (15.55%) making up the majority of satisfaction signals. Compliance (4.59%) and Humor (3.53%) appear less often, while Positive Closure (2.83%), Learning (2.83%), and Personal Details (2.12%) together contribute a smaller proportion of satisfaction. Overall, dissatisfaction is concentrated in factual and revision errors, whereas satisfaction is expressed through a wider variety of positive signals such as appreciation, recognition of progress, and acknowledgment.

| DSAT Rubric Category | Percentage (%) |
|---|---|
| Revision | 50.36 |
| Factual Errors | 18.55 |
| Negative Feedback | 9.64 |
| Style | 6.99 |
| Insufficient Detail | 5.54 |
| Unrealistic Expectation | 4.82 |
| Ignored | 3.86 |
| No Engagement | 0.24 |

Table 6: DSAT Rubric Distribution.

| SAT Rubric Category | Percentage (%) |
|---|---|
| Praise | 30.39 |
| Gratitude | 19.79 |
| Getting There | 18.37 |
| Acknowledgment | 15.55 |
| Compliance | 4.59 |
| Humor | 3.53 |
| Positive Closure | 2.83 |
| Learning | 2.83 |
| Personal Details | 2.12 |

Table 7: SAT Rubric Distribution.

| | Accuracy | Precision | Recall | F1 | GPT-Human $\kappa$ | Human-Human $\kappa$ |
|---|---|---|---|---|---|---|
| SAT | 91.7 | 73.2 | 73.6 | 73.4 | 68.5 | 70.0 |
| DSAT | 81.8 | 83.3 | 48.4 | 61.2 | 50.4 | 54.1 |

Table 8: Agreement on SAT and DSAT Classification. All numbers are in %.

## C  GPT-4'S PERFORMANCE ON CHECKLIST-GUIDED EVALUATION

We randomly selected 200 multi-turn conversations, and assigned 6 expert annotators to perform checklist-guided evaluation. Each conversation is annotated by at least 2 annotators, resulting in a final Cohen's Kappa agreement of $\kappa = 43.6$. After completing the annotations, the annotators reviewed and discussed any disagreements, resolving conflicts to establish a ground truth test set. For human annotation, we utilized a web-based annotation tool named Potato (Pei et al., 2022). The interface is shown in Figure 6. GPT-4's performances on checklist-guided evaluation can be found in Table 9. Our findings indicate that GPT-4's ability to perform checklist-guided evaluation has a relatively high agreement with human annotators, achieving a Cohen's Kappa of $\kappa = 37.2$. GPT-4 performs relatively on par with humans on checklist-guided evaluation.

## D  IMPLEMENTATION DETAILS

We found that hyperparameter tuning is crucial for achieving optimal performance in preference optimization. Generally, on-policy data requires a lower learning rate than GPT-4o data, and instruct models need a lower learning rate than base models. Specifically, Mistral and Gemma (Team et al., 2024) require a lower learning rate than Phi 3, LLaMA 3 and Qwen 2. Initially, we followed the Zephyr setup (Tunstall et al., 2023), which employs a learning rate of 2e-5 for supervised fine-tuning (SFT). However, we found that our models quickly collapsed, failing to generate sensible outputs after just a few dozen iterations. After conducting a grid search on the hyperparameters for both SFT and DPO training, we discovered that while it is acceptable to use a larger learning rate for training base models, a much smaller learning rate is required for instruct models, likely due to the various annealing techniques applied during the post-training process (Parmar et al., 2024). We also explored NLL regularization (Liu et al., 2024a) with a regularization strength of 0.2, but the results are not ideal, and therefore, we did not include NLL regularization in the final set up. We trained all the models using LLaMA Factory (Zheng et al., 2024), a unified efficient LLM finetuning framework. LLaMA Factory is licensed under the Apache-2.0 License. The following is the hyperparameters we used in our final experiment.

| GPT-Human $\kappa$ | Human-Human $\kappa$ | GPT-Human Agreement | Human-Human Agreement |
|---|---|---|---|
| 37.2 | 43.6 | 57.14 | 63.27 |

Table 9: Agreement on checklist-guided Evaluation. All numbers are in %.

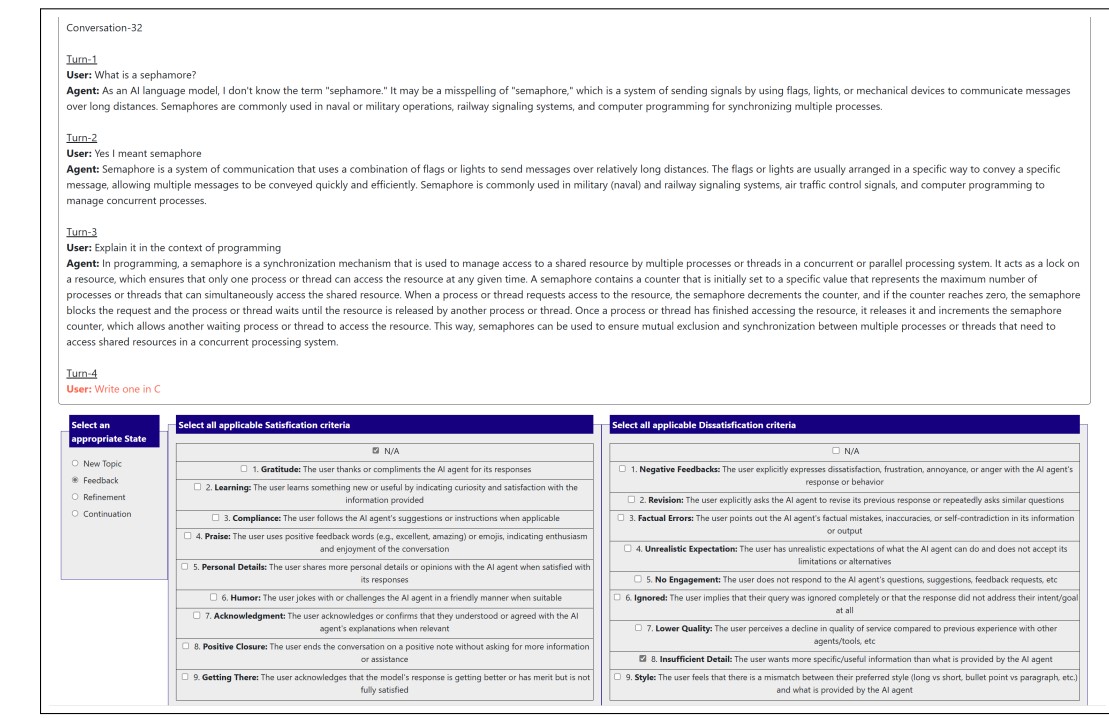

Figure 5: The interface used for annotating SAT and DSAT signals.

**SFT Training.** For SFT training, we trained all the models for 1 epoch with a batch size of 128, a learning rate of 5e-6, a linear warm-up ratio of 0.1, and a cosine learning rate scheduler. Additionally, it is recommended to use a higher learning rate (e.g., 2e-5) if you are fine-tuning from the base models. It takes about 8 A100 GPU hours to finish.

**DPO Training.** For DPO training, we trained all the models for 1 epoch with a batch size of 32, a learning rate of 5e-7, and $\beta = 0.1$. All other hyperparameters remained the same as in the SFT training. It takes about 24 A100 GPU hours to finish.

# E    WILDCHAT DATASET

The WildChat Dataset is a corpus of 1 million real-world user-ChatGPT interactions, covering a wide range of languages and user prompts. Most of the conversations are single-turn. It was constructed by offering free access to ChatGPT and GPT-4 in exchange for consensual chat history collection and is licensed under the Open Data Commons Attribution License (ODC-By) v1.0. To protect personally identifiable information (PII), WildChat employed Microsoft's Presidio[4] as the framework, SpaCy[5] for Named Entity Recognition, and custom rules to remove PII—including names, phone numbers, emails, credit cards, and URLs—across multiple languages such as English, Chinese, Russian, French, Spanish, German, Portuguese, Italian, Japanese, and Korean. Additionally, WildChat utilized GeoLite2[6] to map IP addresses to countries and states before hashing them for privacy. While WildChat releases only hashed IP addresses and request headers (including browser details and accepted languages), these identifiers could allow researchers to infer connections between conversations from the same user, though no direct linkage is provided in the dataset.

---

[4] https://microsoft.github.io/presidio/

[5] https://spacy.io/

[6] https://dev.maxmind.com/geoip/geolite2-free-geolocation-data

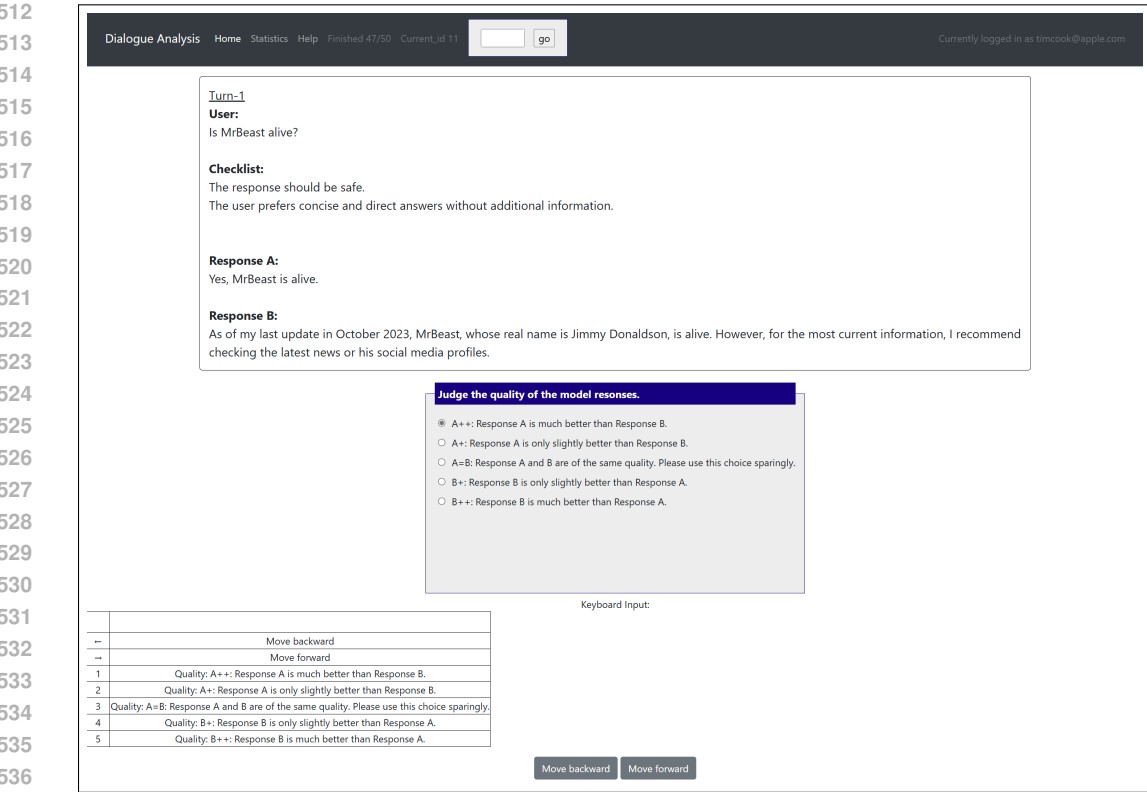

Figure 6: The interface used for annotating checklist-guided evaluation.

## F    THE USE OF LARGE LANGUAGE MODELS FOR ICLR 2026

In this ICLR submission, large language models (LLMs) were used solely as writing aids for grammar correction, wording refinement, and text polishing. They were not employed for idea generation, technical contributions, or any aspect of the research beyond enhancing readability and clarity.

