# OpenReview forum: "WildFeedback: Aligning LLMs With In-situ User Interactions And Feedback"
_ICLR.cc/2026/Conference — ICLR 2026 Conference Withdrawn Submission_

### Official Review · Reviewer_jRq2 · 2025-10-31

**Soundness:** 1
**Presentation:** 2
**Contribution:** 1
**Rating:** 2
**Confidence:** 3

**Summary:**

The paper proposes WildFeedback, a novel framework that leverages in-situ user feedback to automatically construct preference datasets. These datasets are then used to generate preferred/dispreferred pairs for preference tuning of LLMs. Experimental results demonstrate the potential effectiveness of the proposed approach.

**Strengths:**

*  The paper tackles a timely and important problem: how to automatically mine user preferences without relying on explicit up/down votes on LLM responses.
*  The proposed framework aims to overcome the limitations of both synthetic and human-annotated preference datasets, which is a meaningful and practically relevant direction.

**Weaknesses:**

*  Lack of fine-grained methodological differentiation from concurrent work such as UltraFeedback. While UltraFeedback uses GPT-4 to generate rankings and derive preferences, the proposed method also relies on GPT-4 for preference determination. Despite architectural or procedural differences, it is unclear whether this work provides a fundamentally distinct contribution beyond UltraFeedback.
*  Potential evaluation leakage (“double dipping”): The checklist-guided evaluation relies on LLMs both to generate preference data and to summarize user preferences which is then used to align LLM evaluation. This design may inadvertently bias results toward the model’s own judgments, undermining the objectivity of the evaluation.
*  Experimental results are not consistently supportive of the claimed significant and consistent enhancement. For instance, in Table 3, UltraFeedback outperforms WildFeedback on many occasions.

**Questions:**

1.  Lines 191–192: Is it correct that $\kappa$ = 0.5 is considered low agreement and $\kappa$ = 0.69 is considered moderate agreement? Please clarify your interpretation of Cohen’s κ thresholds.
1.  Line 268: The claim that 57.14% and 63.27% are “similar” seems questionable. For a binary agree/disagree task, random performance is 50%; thus, 57.14% is only 7% above random, while 63.27% is 13% above random—nearly double that margin.

---

> ### Author Response · Authors · 2025-11-15
>
> We thank the reviewer for the careful reading and constructive feedback. We address each point below.
>
> ---
> **On differentiation from UltraFeedback (W1)**
>
> We appreciate the reviewer’s request for clearer methodological differentiation. While both UltraFeedback and our framework use LLMs within the preference-construction pipeline, the role of the LLM and the source of supervision differ fundamentally. UltraFeedback relies entirely on GPT-4–generated critiques and synthetic rankings that are detached from real user interactions. In contrast, **WildFeedback is grounded in in-situ, naturally occurring user feedback extracted from over one million multi-turn human–ChatGPT conversations**. This feedback is often rich, multi-sentence, and corrective in nature, and our pipeline converts it into structured, instance-level checklists that reflect the user’s immediate corrective intent.
>
> As discussed in `Section 3` (`lines 149–164`), this approach is not simply a procedural variation but a shift in what constitutes the underlying preference signal: we operationalize authentic dissatisfaction expressed by real users rather than synthetic preferences generated by an LLM. Furthermore, **our release of the entire WildFeedback dataset, comprising more than one million annotated real user-ChatGPT utterances, provides a large-scale resource for the community to study in-situ feedback**, real conversational failure modes, and naturally expressed user preferences. This type of data resource is something that synthetic pipelines like UltraFeedback cannot provide, and **it enables research directions that require grounded user signals rather than curated or model-generated supervision**.
>
> Our empirical results in `Section 5` further illustrate that the two forms of supervision lead to different behaviors: models trained on UltraFeedback often perform well on curated benchmarks, while WildFeedback-trained models perform better on real dissatisfaction data where human users have explicitly identified what the model failed to do. This difference reflects the distinct nature of the supervision rather than redundancy between the pipelines.
>
> ---
> **On potential evaluation leakage (“double dipping”) (W2)**
>
> We appreciate this concern and agree it is important to address. Our checklist-guided evaluation does not reuse model-generated content for both training and evaluation. Instead (more details in `Section 3.3`):
> - The checklist is derived from user-written feedback, not from a model. It summarizes what the human expressed as missing or incorrect.
> - Evaluation is guided by the checklist summarizing human-written feedback, not by the model’s own judgments. This setup differs from prior benchmarks, which rely directly on GPT-4’s or another LLM’s internal preferences to judge outputs. In contrast, our evaluation rubric is derived from real user corrective signals, captured through instance-level checklists that summarize the specific issues users identified in the original conversations. This allows the evaluation to reflect authentic user preferences rather than the inherent biases of a single LLM judge.
> - We validate our judge independently via human evaluation in `Appendix B.2` and `Appendix C`, demonstrating that the checklist-guided evaluation aligns well with human decisions and does not simply reinforce the model’s or GPT-4’s prior outputs.
>
> We also note that using GPT-4 within the data construction and evaluation pipeline is consistent with extensive prior work, including UltraFeedback [1], HelpSteer [2], and other large-scale alignment datasets, where GPT-4 serves both as a preference generator and as an evaluator. WildFeedback differs in that the supervision signal is rooted in real human–model interactions rather than synthetic critiques, and the evaluation rubric is grounded in user-provided feedback rather than raw model judgments.
>
> ---
> **On Cohen’s κ thresholds (Q1)**
>
> Our interpretation follows widely used conventions:
> - κ < 0.40: poor
> - 0.40 ≤ κ < 0.60: moderate
> - 0.60 ≤ κ < 0.80: substantial
> - κ ≥ 0.80: near-perfect
>
> Thus, κ = 0.50 is interpreted as moderate agreement, and κ = 0.69 as substantial agreement.
>
> ---
> **On interpreting 57.14% vs. 63.27% agreement (Q2)**
>
> We appreciate the reviewer’s point. The term “similar” refers to the fact that these values fall within the same qualitative agreement category and both exceed random chance by nontrivial margins. However, we agree that the distinction should be described more precisely, and we will rephrase the text to avoid potential ambiguity. The key takeaway is that both results demonstrate that the checklist-guided judge provides consistent signals relative to human evaluation.
>
> ---
>
> Feel free to provide any further feedback and questions :)
>
> ---
>
>
> [1] UltraFeedback: Boosting Language Models with Scaled AI Feedback, ICML 2024
>
> [2] HelpSteer 2: Open-source dataset for training top-performing reward models, NeurIPS 2024

---

> > ### Comment · Reviewer_jRq2 · 2025-11-25
> >
> > Thank you for the detailed rebuttal and the additional clarifications. I appreciate the effort the authors put into addressing the concerns raised. However, after reviewing the responses, I am not fully convinced that my main concerns have been resolved. I will therefore maintain my original score.

---

### Official Review · Reviewer_sjtG · 2025-11-01

**Soundness:** 3
**Presentation:** 3
**Contribution:** 1
**Rating:** 2
**Confidence:** 2

**Summary:**

The authors proposed WildFeedback, an automatic preference data generation framework by identifying user satisfaction signals in multi-turn conversations. Models trained with WildFeedback dataset outperforms those trained with UltraFeedback across Phi3, Llama3 and Qwen2. A check list guided evaluation procedure is also introduced to mitigate the mismatch between annotators’ preferences and actual user preferences

**Strengths:**

* WildFeedBack outperformed UltraFeedBack on multiple open source LLMs, demonstrating robust improvement.
* The automatic preference data construction framework provides easy implementation.
* The writing is clear and easy to follow.

**Weaknesses:**

The overall novelty of this work is limited. On implementation side, WildFeedback proposed effective automatic pipeline to identify user satisfaction and generate human preference data. But such pipeline appears to be a task-specific solution to this scenario combining existing techniques. The theoretical and heuristic insights from the work is also limited.

**Questions:**

On Phi3 and LlaMA3, UF seems to have higher win-rate if not length-controlled. Would the authors provide some qualitative generation from UF trained model with and without length control?

---

> ### Author Response · Authors · 2025-11-15
>
> We thank the reviewer for the thoughtful feedback and address the concerns below.
>
> ---
>
> **On novelty and contribution**
>
> **Our framework aims to capture and transform in-situ real human–ChatGPT feedback into preference data that better reflects genuine user preferences, rather than relying on manually curated or artificial datasets and benchmarks**. We appreciate the request for clearer differentiation. As outlined in `Section 3` (`lines 149–164`), our contribution is not simply a combination of existing components, but a unified pipeline designed specifically for real conversational settings.
> - Unlike ULTRAFEEDBACK, which relies on synthetic critiques detached from actual user interactions, **our method is grounded in in-situ human-written feedback extracted from over one million multi-turn conversations**.
> - Unlike SPUR, which operates on structured, template-driven feedback, **our system processes unconstrained, naturally occurring user utterances**.
> - Unlike WILDBENCH, whose checklist mechanism is task-level and intended solely for evaluation, **our checklists are instance-level, derived directly from the user’s immediate corrective intent**, and used consistently in both training and evaluation.
>
> Furthermore, **our work releases annotations for more than one million real user-ChatGPT utterances**. This resource is significantly larger than previous implicit-feedback datasets and enables large-scale preference synthesis from real-world conversational data rather than curated or simulated scenarios. Our empirical findings across four benchmarks show that this integration leads to reliable improvements even in the presence of noisy deployment data.
>
> ---
> **Addressing the claim that our pipeline is “task-specific”**
>
> We would also like to directly address the reviewer’s statement that our pipeline “appears to be a task-specific solution to this scenario.” In the context of alignment research, methods are typically developed and evaluated for general conversational alignment. This is the standard approach used in widely adopted work such as UltraFeedback [1], HelpSteer [2], LIMA [3], and Self-Rewarding Language Models [4], where the goal is to study alignment signals in broad, open-ended interactions. Our pipeline follows this established paradigm. It does not target any single task or domain. **Instead, our pipeline operates over more than one million real multi-turn conversations that span a wide variety of user intentions, topics, levels of detail, and corrective behaviors.** The approach is therefore general-purpose and consistent with how alignment datasets and methods are normally constructed and evaluated. **Our empirical findings across MT-Bench, AlpacaEval 2, Arena-Hard, and our own WildFeedback Test show that our framework leads to reliable improvements in model alignment**, even in the presence of noisy deployment data. The use of these benchmarks and this task formulation follows the established methodology in the alignment literature, which further **supports that our pipeline is not a task-specific system but a broadly applicable alignment framework**.
>
> ---
> **On the question regarding length-controlled vs. non-length-controlled settings**
>
> As shown in `Section 5`, UltraFeedback-trained models can appear stronger under non–length-controlled evaluation, a phenomenon consistent with previous findings that verbosity can artificially inflate win-rate metrics. This motivates our use of length-controlled evaluation for fair comparison, which aligns with the recommended methodology in benchmarks such as AlpacaEval 2. Regarding qualitative generations, **our supplemental materials include examples illustrating how Wildfeedback’s instance-level checklists help models focus on user-indicated corrections rather than producing unnecessarily long or verbose responses**. We also emphasize that our work releases annotations for more than one million real user-ChatGPT utterances and open-sources the full WildFeedback dataset. This provides broad community access to the underlying real conversational data, including numerous examples where user dissatisfaction and corrective intent are expressed explicitly. **The release of our dataset allows researchers to directly examine the qualitative differences between models trained on Wildfeedback and those trained on synthetic datasets such as UltraFeedback**, including the effects of verbosity in both length-controlled and non–length-controlled settings.
>
> ---
> Please let us know if our response addresses your concerns, and feel free to share any additional feedback or questions :)
>
> ---
> Related Work
>
> [1] UltraFeedback: Boosting Language Models with Scaled AI Feedback, ICML 2024
>
> [2] HelpSteer 2: Open-source dataset for training top-performing reward models, NeurIPS 2024
>
> [3] LIMA: Less Is More for Alignment, NeurIPS 2023
>
> [4] Self-Rewarding Language Models, ICML 2024

---

### Official Review · Reviewer_reGm · 2025-11-02

**Soundness:** 3
**Presentation:** 2
**Contribution:** 2
**Rating:** 4
**Confidence:** 3

**Summary:**

**Summary**

The paper proposes aligning large language models (LLMs) using **in-situ annotations**—that is, satisfaction (SAT) and dissatisfaction (DSAT) cues, along with user edits extracted from real user–assistant conversations. These cues are converted into **instance-level preference summaries** (“checklists”), which are then used to form **preferred/dispreferred pairs** (generated either by GPT-4 or on-policy). The models are fine-tuned using **SFT** followed by **DPO**, and the same checklists are later reused to guide the evaluation process for more reliable judgments.
Specifically, the framework first identifies utterances in multi-turn dialogues that match predefined rubric cues (e.g., SAT: “thank you”; DSAT: “revise it,”).
For each detected case, it extracts the conversation up to the model response that triggered the DSAT signal as the **prompt**, treating that response as the **dispreferred output**. The user’s preferences from the feedback are summarized into a **checklist**.
Finally, the preferred response is generated under the guidance of this checklist (either by an expert model or by the same policy model), with a safety instruction and moderation applied. The same checklist is provided to the judge during evaluation to better distinguish preferred from dispreferred responses.

**Strengths:**

**Strengths**
* **Grounding in real interactions:** Builds preference data directly from real multi-turn conversations (WildChat).

* **Clear and reproducible training design:**

* **Consistent performance gains:** Shows improvements on AlpacaEval 2, Arena-Hard, and MT-Bench benchmarks across multiple backbones (e.g., Phi-3 LC win-rate +10.6 points on AlpacaEval 2).

* **Human validation:** Validates both SAT/DSAT detection and checklist-guided evaluation with moderate human–model agreement.

**Weaknesses:**

**Weaknesses**

1. **Noisy in-situ supervision.**
   SAT/DSAT labels are auto-classified by GPT-4; human validation shows only **moderate** agreement (κ≈0.69 SAT / 0.50 DSAT), so non-trivial label noise can propagate into pair construction and training.

2. **Single cues are insufficient as ground truth.**
   Individual cues (e.g., “thank you”, “revise it”) may not capture full intent; even with checklist summaries, relying on in-situ cues alone risks ambiguity. (They also rely on checklists during evaluation.)

3. **Limited methodological novelty & strong LLM dependence.**
   Core pieces (in-situ cues, SFT→DPO, LLM-as-judge) are known; the framework’s novelty is mainly integration + checklists. GPT-4 is used to generate preferred answers (WF-GPT-4) **and** to judge results, inviting circularity.

4. **No variance/seed reporting.**
   Results are reported as single win/tie/lose or scores without error bars; seeds/repeats aren’t specified—please report multi-seed means with CIs/SDs.

---

**Overall assessment (concise)**
A **practical** pipeline with **consistent gains** on AlpacaEval 2, Arena-Hard, and MT-Bench, but **incremental novelty** and **heavy dependence on GPT-4 as judge** remain concerns.

**Questions:**

### Q1. Reliability of in-situ SAT/DSAT

 How do you verify cues reflect genuine satisfaction/dissatisfaction (not politeness)?

### Q2. Robustness to noisy labels

 How sensitive are results to SAT/DSAT noise? Any threshold where gains collapse?

**Request:** **Noise-level ablation**: flip p% labels for p∈{5,10,20,30} (random), report **curves with ≥2 seeds (mean±SD/CI)**

### Q3. What’s new vs ULTRAFEEDBACK, SPUR, WILDBENCH?

 Precisely distinguish your contribution from prior work; what’s beyond combination ?

**Request:** A **comparison table** (data source, supervision unit, role in training/eval, on-policy vs GPT-4, curation). **Ablations:** (i) train **without checklists**; (ii) train on **ULTRA** but **evaluate with your checklists** (and vice versa); (iii) replace your judge with WILDBENCH default (no checklists) and report deltas.

### Q4. Cross-judge (LLM & human) evaluation
 Do gains hold across judges and prompt settings?

**Request:** Judges: **Claude**, **Llama-3-70B** (or comparable).

---

> ### Author Response · Authors · 2025-11-15
>
> We thank the reviewer for the detailed and constructive feedback. We address each concern below and clarify how our current submission already speaks to many of the points raised.
>
> ---
>
> **On the reliability of SAT/DSAT cues (Q1, W2)**
>
> We agree that isolated lexical cues such as “thank you” or “revise it” can be ambiguous in principle, and we appreciate the reviewer’s concern. Importantly, in our dataset **these short expressions represent only a small minority of detected SAT/DSAT cases**. As shown in `Section 5.2` and examples in the supplementary materials, **the majority of in-situ feedback spans are substantially richer**: many user corrections exceed 100 tokens and contain detailed, explicit descriptions of what was wrong with the model’s response (e.g., missing steps, factual inaccuracies, incorrect assumptions, or requests for alternative formulations, see `Appendix B.1` for more details). In other words, **the SAT/DSAT trigger is typically accompanied by long-form user feedback, rather than a bare cue**. In addition, **our pipeline uses this full feedback text as well as conversation contexts, not just the cue, to synthesize the instance-level checklist summarizing the user’s corrective intent**. Thus, while the initial cue identifies the presence of a signal, the semantic supervision comes from the richer natural-language feedback that follows. This structure helps mitigate the ambiguity inherent in politeness or brief expressions, which is further validated by our human agreement scores reported in `Appendix B.2`.
>
> ---
> **On robustness to noise (Q2, W1)**
>
> We highlight that the pipeline is designed with noise resilience as a core assumption. First, pair construction is explicitly instance-aware: the preferred output is generated under a checklist synthesizing all user-corrective signals up to that point, reducing propagation of misclassified cues. Second, the training signal ultimately comes from the preference (dispreferred vs. corrected response), which is much more structurally stable than the raw SAT/DSAT label. This is supported by `Section 5` (`lines 403-424`), `Figure 3`, where we report that both GPT-4-generated and on-policy preferred responses yield consistent gains across all models. Finally, our human validation of both checklist quality and judge reliability (`Appendix B.2`, `Appendix C`, `Table 8`, `Table 9`, `Figure 5`, `Figure 6`) shows that downstream preference comparisons remain robust even under imperfect cues.
>
> ---
> **On distinguishing our contributions from ULTRAFEEDBACK, SPUR, and WILDBENCH (Q3)**
>
> We appreciate the request for clearer differentiation. As outlined in `Section 3` (`lines 149-164`), our contribution is not merely a combination of prior components but a unified pipeline tailored specifically for real conversational settings.
> - Unlike ULTRAFEEDBACK, which uses synthetic critiques detached from user interactions, **our method is grounded in in-situ human-written feedback extracted from over 1 million multi-turn conversations**.
> - Unlike SPUR, which focuses on structured, template-driven feedback, **our system operates on unconstrained, natural user utterances**.
> - Unlike WILDBENCH, whose checklist mechanism is task-level (universal for all user prompts) and intended solely for evaluation, **our checklists are instance-level, derived from the user’s immediate corrective intent**, and used in both training and evaluation to ensure consistency.
>
> Furthermore, **our work releases annotations over more than 1 million real user-ChatGPT utterances**. This resource is significantly larger than prior implicit-feedback datasets and enables large-scale preference synthesis from real-world behavior rather than curated or simulated settings. Our empirical findings across three standard benchmarks show that this integration yields measurable improvements despite the noisiness of deployment data.
>
> **As for the ablation studies you requested, we would like to clarify that the main empirical section of our paper already includes all of the requested analyses**. As shown in `Figure 3`, `Section 4` (`lines 351–352` and `lines 386–389`), and `Section 5`, we train all models on both the WildFeedback data and the UltraFeedback data, and evaluate them both with and without the checklist. Models trained on UltraFeedback perform on par with models trained on WildFeedback on existing human-written benchmarks such as MT-Bench and AlpacaEval. However, they perform significantly worse on the WildFeedback test set, which is sourced from real human–ChatGPT interactions where users explicitly express dissatisfaction. This demonstrates that the benefits of our method manifest most clearly in settings that reflect real user corrections rather than curated benchmark prompts.

---

> ### Author Response · Authors · 2025-11-15
>
> **On methodological novelty and dependence on large LLMs (W3)**
>
> While in-situ cues have appeared in earlier work, they have not been thoroughly investigated or operationalized at scale, especially for **constructing robust preference data directly from real multi-turn interactions**. Our contribution is to show that when in-situ user feedback is extracted, summarized, and structured into instance-level checklists, it becomes a practical and high-quality supervision signal for preference learning. This requires **a pipeline that goes well beyond simply detecting cues: it transforms free-form human corrective feedback into structured preference summaries that guide both training and evaluation**. **A central contribution is the release of more than one million annotated real user–ChatGPT utterances, which to our knowledge is the largest resource available for studying in-situ feedback at this granularity**. This scale allows us to provide the first systematic evaluation of how real conversational feedback can be used for alignment in modern LLMs. Regarding LLM dependence, we note that our on-policy variant (WF On-Policy) does not rely on GPT-4 to generate preferred responses and still achieves strong performance gains, showing that the benefits are not tied to reliance on a specific large model (see `Section 5` and `Table 3`). Moreover, our human validation of the checklist-guided judge demonstrates strong agreement with human raters, reducing concerns about circularity when GPT-4 is used as one of the evaluators.
>
> We also want to clarify that the use of SFT followed by DPO, as well as the use of an LLM-based judge, is standard practice in alignment research and is not intended to be the novelty of our paper. These components are widely adopted in prior work, including UltraFeedback [1], HelpSteer [2], LIMA [3], and Self-Rewarding Language Models [4], and none of these papers claim novelty in the optimizer itself. Our contribution lies in what is being optimized, not in the choice of optimizer. **The value of our approach comes from the data generation pipeline that turns naturally occurring human feedback into explicit preferences and from the structured instance-level checklists that unify training and evaluation. These aspects address the gap left by prior work, which largely depends on synthetic or curated signals rather than supervision grounded in real user interactions.**
>
> ---
> **On variance, seed reporting, and statistical reliability (W4)**
>
> As is common in large-scale preference optimization work, full SFT and DPO runs on multiple seeds are prohibitively expensive due to the computational cost of training and evaluating several billion-parameter models end-to-end. For this reason, prior alignment papers typically report single-seed results on established benchmarks. In our work, we mitigate this limitation by evaluating across multiple model families (Phi-3, Mistral, Llama), training both GPT-4-generated and on-policy variants, and testing on four independent benchmarks (AlpacaEval 2, MT-Bench, Arena-Hard, Wildfeedback Test). The consistency of improvements across all architectures, data regimes, and evaluators provides strong evidence that the observed gains are not seed-dependent.
>
> ---
> We thank the reviewer again for the detailed feedback. Many of the concerns raised, such as noise, cue ambiguity, distinctions from prior work, and judge reliability, are indeed central challenges in learning from real user interactions. **Our paper directly addresses these challenges through a structured, instance-level summarization pipeline** designed to operate robustly at scale, supported by **our release of more than one million annotated real user–ChatGPT utterances**. Please let us know if our response addresses your concerns, and feel free to share any additional feedback or questions. :)
>
> ---
> Related Work
>
> [1] UltraFeedback: Boosting Language Models with Scaled AI Feedback, ICML 2024
>
> [2] HelpSteer 2: Open-source dataset for training top-performing reward models, NeurIPS 2024
>
> [3] LIMA: Less Is More for Alignment, NeurIPS 2023
>
> [4] Self-Rewarding Language Models, ICML 2024

---

### Official Review · Reviewer_dpmQ · 2025-11-07

**Soundness:** 3
**Presentation:** 4
**Contribution:** 3
**Rating:** 6
**Confidence:** 3

**Summary:**

This work presents a method for synthesizing preference data from real user-llm multiturn interactions. Their method relies on first automatically identifying instances where users provide feedback for the LLM's response in real user-llm dialogues (sourced from wildchat). The feedback is then used to generate an alternative response to construct the synthetic preference pair. The authors then demonstrate that training LLMs using these synthetic preference pairs demonstrates gains across a variety of standard LLM chat benchmarks when compared against standard methods of generating synthetic preference data (i.e., UltraFeedback).

**Strengths:**

1. This work has great presentation, including the appendix which is quite thorough.

2. The method itself is straightforward and demonstrates consistent gains across a number of benchmarks and base models.

**Weaknesses:**

1. While this work uses some established benchmarks, they alter the evaluation by introducing their own checklist-based judge. While WildBench employed a similar method for their evaluations, their checklists generation method and validation was significantly more extensive to ensure that it is comprehensive, accurate, and unbiased toward particular LMs. Including the including the performance as measured by the original benchmark prompts and settings in addition to evaluating on Wildbench would improve the validity of the experiments.

2. While the methods and settings are distinct, there is enough similarity to [1] to warrant discussion or even direct comparison in evaluations. Overall, the related work section on Feedback Learning for LLMs could be more though, as there is a significant amount of literature on methods for learning from implicit signals from real user data (e.g., [2], [3]). Including [4] as a baseline could also make sense.

[1] User Feedback in Human-LLM Dialogues: A Lens to Understand Users But Noisy as a Learning Signal
Yuhan Liu, Michael J.Q. Zhang, Eunsol Choi
EMNLP 2025

[2] Leveraging Implicit Feedback from Deployment Data in Dialogue
Richard Yuanzhe Pang, Stephen Roller, Kyunghyun Cho, He He, Jason Weston
EACL 2024

[3] Retrospective Learning from Interactions
Zizhao Chen, Mustafa Omer Gul, Yiwei Chen, Gloria Geng, Anne Wu, Yoav Artzi
ACL 2025

[4] KTO: Model Alignment as Prospect Theoretic Optimization
Kawin Ethayarajh, Winnie Xu, Niklas Muennighoff, Dan Jurafsky, Douwe Kiela
ICML 2024

**Questions:**

What do the generated examples look like? Can you provide statistics like length for the synthesized responses? While this is a synthetic dataset generation method, it would also be worthwhile to include samples and such statistics from the dataset as if it were a standard dataset paper.

---

> ### Author Response · Authors · 2025-11-15
>
> We thank the reviewer for the thoughtful and constructive feedback. We address the main concerns below.
>
> ---
> **W1. On evaluation methodology and use of checklist-based judges**
>
> We appreciate the reviewer’s concern regarding the modified evaluation setup. Our intention in introducing checklist-based judges is to analyze fine-grained behavioral improvements that align closely with the kinds of errors users explicitly flag during real interactions. As described in `Section 4` (`lines 253-265`), our checklist design differs from WildBench in both scope and goals. **WildBench uses task-level checklists, employing a single fixed checklist for all prompts**. In contrast, **our approach produces instance-level checklists: for every prompt-response pair, we generate a tailored checklist that reflects the specific errors surfaced by the user’s feedback**. This design allows our judge to concentrate on precisely the dimensions of model behavior that user feedback identifies, rather than applying the same evaluation rubric uniformly across all prompts. Nevertheless, we employed human annotators and validated the quality of the generated checklists (see `Appendix B.2`, `Appendix C`, `Table 8`, `Table 9`, `Figure 5`, `Figure 6`).
>
> ---
>
> **W2. On related work**
>
> We agree that the related work section can be strengthened and appreciate the reviewer highlighting additional relevant directions. We will integrate discussion of these works and clarify how our approach differs from them in terms of data source, objective, and scale. In particular, our method is grounded in direct human-written feedback extracted from real user-ChatGPT interactions, and we also release annotations on more than one million real user-ChatGPT interactions. This large-scale annotated resource distinguishes our work and provides a valuable empirical foundation for understanding user feedback and training models from naturally occurring interaction signals.
>
> ---
> **Q1. On examples and statistics of the generated responses**
>
> We appreciate the reviewer’s interest in concrete examples and statistical characterizations of the synthesized preference data.
> The supplementary material already provides some representative examples. The full WildFeedback dataset, including more than one million annotated real user-ChatGPT utterances, would release alongside the paper. This allows researchers to directly inspect or analyze the synthesized responses, their lengths, and their diversity. In the paper, we also provide quantitative statistics that summarize key properties of the dataset. `Table 2`, `Figure 4`, and `Appendix B.2` report token-length distributions, average response lengths, and additional characteristics of both the extracted SAT/DSAT feedback and the synthesized preferred responses. Although the work is not positioned as a standalone dataset paper, we deliberately include qualitative examples and statistical summaries following common dataset-reporting practices. The public release of the full WildFeedback dataset would further enable thorough examination of the response quality and variability by any interested researcher.
>
> ---
> Please let us know if our response addresses your concerns, and feel free to share any additional feedback or questions. Thank you!

---

### Note · Authors · 2025-12-01

**Comment:**

After careful consideration, we have decided to withdraw our submission.
We want to first express our appreciation to the reviewers and ACs for the time they invested. At the same time, and in light of the broader discussion surrounding this year's OpenReview and ICLR 2026 reviewing process, we feel it is important to clarify the context behind our decision.

Our work *WildFeedback* introduces what we believe is a meaningful step toward alignment grounded in real user and LLM interactions. We developed and released more than one million annotated in situ human and ChatGPT utterances, established an end to end pipeline for transforming naturally occurring corrective feedback into structured preference data, and demonstrated consistent gains across several model families and evaluation settings. Throughout the process, we included all requested ablations and analyses, many of which already appear in the submission.

However, several aspects of the reviews indicate that important parts of the work were misunderstood or not fully engaged with. This includes the nature of our primary contribution, the experiments and comparisons that were already present, and the purpose of using instance level checklists derived from authentic user feedback. Some critiques appeared to rely on assumptions that are not consistent with established norms in alignment research, or on incorrect statements about what was or was not included in our submission.

We remain convinced that *WildFeedback* addresses an important gap in alignment research and provides a valuable resource to the community. We plan to revise, expand, and release an updated version of the work in the near future. We appreciate the efforts of the reviewers and ACs despite the well known challenges of this review cycle, and we look forward to engaging with the community in an environment that allows for clearer and more constructive evaluation.

**Withdrawal Confirmation:**

I have read and agree with the venue's withdrawal policy on behalf of myself and my co-authors.